# Learning from Self Critique and Refinement for Faithful LLM Summarization

## Abstract

Large Language Models (LLMs) often suffer from hallucinations: output content that is not grounded in the input context, when performing long-form text generation tasks such as summarization. Prior works have shown that hallucinations can be be reduced by iteratively critiquing and refining previously generated outputs using either the same model or a more powerful teacher model as the critique. However, these approaches either require additional test-time compute or assume access to more powerful teacher models, making them costly and less practical. In this work, we propose Self Critique and Refinement-based Preference Optimization (SCRPO), which is a self-supervised training framework that first constructs a preference dataset by leveraging the LLM's own critique and refinement capabilities, and then applies preference learning to improve the same LLM for faithful summarization. Experiments on three summarization benchmarks demonstrate that our approach outperforms state-of-the-art self-supervised learning methods in terms of faithfulness metrics while either maintaining or improving other metrics that measure the overall quality of the summary. Moreover, compared to test-time refinement, our approach not only improves efficiency but also results in more faithful summaries.

## 1 Introcution

Abstractive summarization refers to the task of producing concise summaries through interpretation and rephrasing of the source document. Recent advances in Large Language Models (LLMs) have led to remarkable performance on this task (Lewis et al. (2020); Zhang et al. (2020; 2024a); Goyal et al. (2022)). However, LLMs still suffer from hallucinations, where part of the generated summary is not supported by the input document (Maynez et al. (2020); Huang et al. (2021)). Though various existing works have sought to enhance the faithfulness of generated summaries, the issue of hallucinations remains unresolved.(Chen et al., 2022; Wan et al., 2024; Wadhwa et al., 2024; Li et al., 2024a; Wan et al., 2025b).

LLMs demonstrate a broad spectrum of abilities, including critique and refinement (Madaan et al., 2023; Chiang & Lee, 2023; Yu et al., 2025). Recent works have explored leveraging these abilities to mitigate hallucinations in abstractive summarization (Wan et al. (2024); Wadhwa et al. (2024); Hu et al. (2024); Wan et al. (2025a)). While these approaches demonstrate effectiveness, they suffer from two notable limitations. First, they often depend on either strong teacher models or multiple specialized models to construct the refinement pipeline. Second, they typically incur additional computational cost at inference time. These limitations make previous approaches less suitable for real-world applications.

In this work, we propose Self Critique and Refinement-based Preference Optimization (SCRPO), a self-supervised training framework that leverages an LLM's self-critique and self-refinement abilities to enhance its own performance in faithful summarization. Given an unlabeled document, we first generate a set of initial summaries using a pretrained LLM. The same LLM is then prompted to critique these summaries with respect to faithfulness, and based on the critique responses, the same LLM is prompted again to refine the initial summaries. The initial and refined summaries are subsequently organized into a preference tuple, where the chosen summary is selected from the refined summary set and the rejected summary is selected from the initial summary set. Through this process, we construct a preference dataset that captures the model's internal knowledge of faith-

fulness. Finally, the same LLM is trained on this dataset, effectively learning to perform faithful summarization from its own self-generated preferences. The resulting LLM incurs no additional inference cost.

We investigate two strategies for the LLM critique component in the proposed SCRPO framework. (1) *Critique with binary feedback:* The LLM is prompted to output a simple yes/no response indicating whether the generated summary has any hallucinated content. (2) *Critique with fine-grained feedback:* Inspired by recent advances in hallucination detection, we decompose the critique process into three steps. Specifically, the LLM is first prompted to extract a set of atomic facts from the summary. Then, it is prompted to verify whether each fact is entailed by the source document. Finally, the non-entailed subset of facts are used to provide a fine-grained feedback.

The proposed SCRPO framework aims at enhancing LLM summarization faithfulness by leveraging the self-critique and self-refinement abilities of the model during training time. Alternatively, these same abilities can also be used directly at inference time to improve the faithfulness of the generated summaries. This naturally raises a key research question: Since SCRPO can be considered as distillation of inference-time refinement, will it reach the performance achievable with refining at inference time? In this paper, we show that SCRPO not only reaches but significantly outperforms its inference-time counterpart in terms of faithfulness, while requiring less inference-time compute. We attribute this to SCRPO's ability to aggregate the LLM's internal knowledge elicited from a broad set of training documents, in contrast to inference-time refinement that summarizes each test document independently.

We evaluate the effectiveness of the SCRPO framework through extensive experiments on three benchmark datasets: XSum and CNNDM which are news summarization datasets, and SAMSum. which is a dialogue summarization dataset. Our results show that SCRPO consistently outperforms previous state-of-the-art self-supervised methods in terms of faithfulness metrics (MiniCheck (Tang et al., 2024) and GPT4-Likert score (Li et al., 2024b)) on all datasets, while either maintaining or improving the overall quality of the summaries measured by GEval scores (Liu et al., 2023).

**Major contributions:**

- We introduce SCRPO, which is a self-supervised training framework that leverages the self-critique and self-refinement abilities of LLMs to construct a preference dataset that improves summarization faithfulness. This framework enables a model to self-improve by learning from its own internal knowledge of faithfulness without any external supervision.

- We demonstrate that SCRPO significantly outperforms its inference-time counterpart in terms of faithfulness, while requiring less compute at inference time.

- Through extensive experiments on three benchmark datasets, we demonstrate that SCRPO outperforms prior state-of-the-art methods in terms of faithfulness metrics, while also either preserving or improving the overall quality of the summaries.

## 2    RELATED WORK

The refinement capability of LLMs has been extensively studied and applied across a variety of tasks, including faithful summarization. Several prior works have leveraged refinement to improve summarization quality. For instance, Wadhwa et al. (2024) fine-tuned an LLM to perform refinement based on the natural language feedback about unfaithful content. Wan et al. (2025a) proposed a multi-agent, multi-model collaboration framework that consists of detection, critique, and reranking steps. Wan et al. (2024) further refined candidate summaries with fine-grained feedback at the level of atomic, non-decomposable facts. While effective, these approaches typically rely on stronger teacher LLMs or external specialized models to build the refinement pipeline, and they also incur additional inference-time computational cost. In contrast, our SCRPO framework distills the internal knowledge of a single LLM through preference data construction, improving the faithfulness of the same model without introducing extra computation at the inference phase.

Previous work has explored improving the faithfulness of LLM summarization without relying on external knowledge or teacher models. One line of research focuses on designing advanced decoding mechanisms that adjust next-token probabilities according to specific criteria. For example, van der

Poel et al. (2022) penalize ungrounded tokens using a context-less model when the next-token distribution has high entropy. Shi et al. (2024) reduce token probabilities through a context-less model controlled by a scaling factor. King et al. (2022) propose a rule-based token-level faithfulness estimator, and constrain the beam search decoding to include only faithful tokens. Another line of work constructs self-generated synthetic data to fine-tune the same LLM. For instance, Choi et al. (2024) create preference datasets by contrasting outputs from decoding strategies of varying quality, and Duong et al. (2025) generate unfaithful responses with a context-less model to build preference data. Compared with these approaches, our SCRPO framework also employs preference learning on self-generated data, but differs in that its preference construction strategy explicitly leverages the LLM's internal capabilities, critique and refinement, resulting in significant improvements over strong pretrained LLMs and prior state-of-the-art methods.

One of the mainstream approaches to hallucination detection relies on fine-grained atomic analysis (Min et al., 2023; Scirè et al., 2024; Yang et al., 2024; Wan et al., 2024; Song et al., 2024; Oh et al., 2025). These methods decompose an LLM response into a set of atomic facts, verify the faithfulness of each fact, and then aggregate the fine-grained verifications to determine whether hallucination is present. Metropolitansky & Larson (2025) further introduce an evaluation method focusing solely on the atomic fact extraction step. This line of work inspires the design of the fine-grained feedback strategy for LLM critique in the SCRPO framework. Unlike prior approaches, we prompt a single LLM to perform both the extraction and verification steps, leverage the fine-grained results for preference data construction, and ultimately train the same LLM on this dataset to achieve faithful summarization.

LLM self-improvement through preference learning or reinforcement learning has become a rapidly growing research direction. Most prior works leverage the self-critique/rewarding capability of LLMs to either train a reward model for reinforcement learning or derive preference labels for self-generated responses. (Yuan et al., 2024; Zhang et al., 2024b; Wang et al., 2024) Bai et al. (2022) and Dong et al. (2025) explore the use of LLM refinement or rewriting as a form of self-improvement. Specifically, Bai et al. (2022) employ refinement during supervised training, whereas Dong et al. (2025) use refinement as a tool to align the response distribution with a target dataset. In contrast, our method integrates both self-critique with numerical and fine-grained feedback, and self-refinement to construct preference datasets. Moreover, our framework does not rely on human-annotated responses (summaries).

## 3 SELF CRITIQUE AND REFINEMENT-BASED PREFERENCE OPTIMIZATION

### 3.1 OVERVIEW

Figure 1 provides an overview of the proposed SCRPO framework, which is a self-improvement-based training mechanism designed to enhance the faithfulness of LLM summarization. Given a set of unlabeled documents $D = \{x\}$ from a specific target domain, our goal is to improve a pre-trained LLM $\pi$ for faithful summarization within this domain. To achieve this goal, we construct a preference dataset $D_{pref} = \{(x, y_{chosen}, y_{rejected})\}$ and employ preference learning to fine-tune $\pi$ with a low rank adapter $\theta$. In the SCRPO framework, preference data construction relies on the self-critique and self-refinement abilities of the same LLM $\pi$. Specifically, for each target domain document $x$, we create a preference triplet by following these four steps: (i) *LLM summarization* - Generate an initial summary $\hat{y} \sim \pi(.|x)$, (ii) *LLM critique* - Critique the faithfulness of the initial summary $\hat{y}$ using $\pi$ to obtain a hallucination score $s$ and a textual feedback $c$ about hallucinations, (iii) *LLM refinement* - If $\hat{y}$ is not faithful (determined by the criterion $s > 0$), use $\pi$ to refine it based on the feedback $c$ to obtain a refined summary $\hat{y}^r$. We repeat these three steps $N$ times, and record all unfaithful initial summaries $\hat{y}$ along with their hallucination scores $s$ and the corresponding refined summaries $\hat{y}_r$. (iv) *Preference triplet selection* - To form a preference triplet, we select the refined summary $\hat{y}_r$ derived from the initial unfaithful summary with the lowest hallucination score as the chosen response $y_{chosen}$, and the initial unfaithful summary $\hat{y}$ with the highest hallucination score as the rejected response $y_{rejected}$. Algorithm 1 demonstrates the full process of preference data construction in our SCRPO framework.

SCRPO extracts the knowledge about faithfulness from $\pi$ in the form of preference data, and incorporates it into the summarization capability of $\pi$ through preference learning, effectively mitigating hallucinations in a self-supervised manner without the need for external resources, stronger teacher

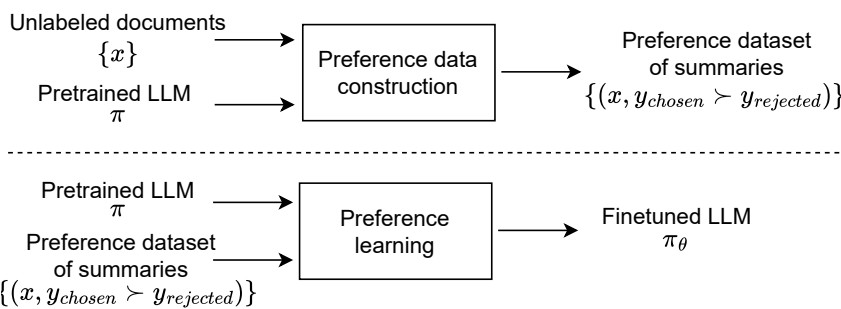

Figure 1: Overview of the proposed SCRPO framework. Given a set of unlabeled documents $\{x\}$, we use a pretrained LLM $\pi$ to construct a preference dataset of summaries, and then finetune the pretrained LLM with preference learning to improve the faithfulness of generated summaries. The details of preference data construction are elaborated in Algorithm 1
.

models, or multi-model pipelines. Also, SCRPO is a training-time approach that introduces no additional computational overhead during inference, making it practical for real-world applications.

The LLM's critique and refinement steps can also be applied directly at inference time to improve the faithfulness of generated summaries. Later in the experiments section, we show that, despite using more compute, directly performing refinement at inference time performs poorly when compared to the proposed SCPRO training approach.

---

**Algorithm 1** Preference data construction in SCRPO framework

---

**Input:** $D = \{x\}$ - unlabeled documents, $\pi$ - pretrained LLM,
  $p_{summ}$ - summarization prompt, $p_{refine}$ - refinement prompt, $N$ - sample size
**Output:** $D_{pref} = \{(x, y_{chosen}, y_{rejected})\}$ - preference dataset
  $D_{pref} \leftarrow \emptyset$
  **for all** $x \in D$ **do**
    $L_{init}, L_{refine}, L_{score} \leftarrow [\,], [\,], [\,]$       ▷ initial/refined summary, and hallucination score lists
    **for** $i \in \{0, 1, \ldots, N-1\}$ **do**
      $\hat{y}^{(i)} \sim \pi(.|x, p_{summ})$       ▷ 1. LLM Summarization
      $(s^{(i)}, c^{(i)}) \leftarrow LLM\_Critique(\pi, x, \hat{y}^{(i)})$       ▷ 2. LLM Critique
      **if** $s_i > 0$ **then**
        $\hat{y}_r^{(i)} \sim \pi(.|x, \hat{y}^{(i)}, c^{(i)}, p_{refine})$       ▷ 3. LLM Refinement
        $(L_{init}, L_{refine}, L_{score}) \leftarrow (L_{init} + [\hat{y}^{(i)}],\ L_{refine} + [\hat{y}_r^{(i)}],\ L_{score} + [s^{(i)}])$
      **end if**
    **end for**
    $y_{chosen}, y_{rejected} \leftarrow$ PREF_TRIPLET_SELECTION$(L_{init}, L_{refine}, L_{score})$
    $D_{pref} \leftarrow D_{pref} \cup \{(x, y_{chosen}, y_{rejected})\}$
  **end for**
  **return** $D_{pref}$

  **function** PREF_TRIPLET_SELECTION$(L_{init}, L_{refine}, L_{score})$       ▷ 4. Preference triplet selection
    $i_{max}, i_{min} \leftarrow \arg\max L_{score}, \arg\min L_{score}$
    $y_{chosen}, y_{rejected} \leftarrow L_{refine}[i_{min}], L_{init}[i_{max}]$
    **return** $y_{chosen}, y_{rejected}$
  **end function**

---

### 3.2 LLM CRITIQUE

LLM critique is responsible for identifying the unfaithful content in the initial summary $\hat{y}$, providing a hallucination score $s$ and a textual feedback $c$. The hallucination score $s$ is used to determine if

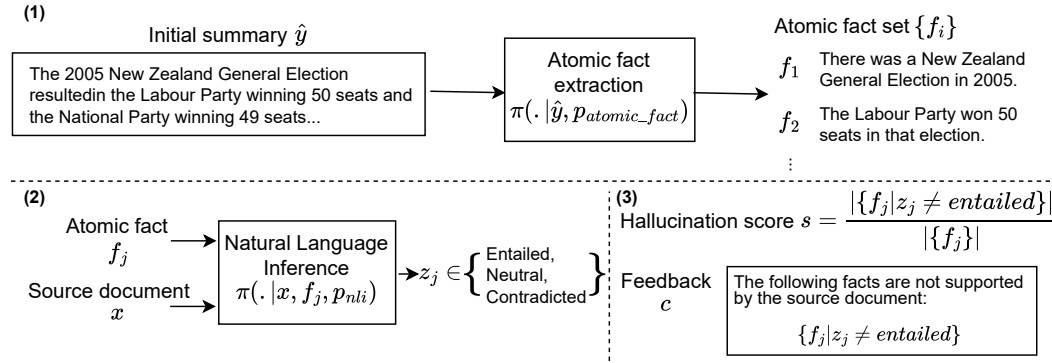

Figure 2: LLM critique with fine-grained feedback. We prompt the same LLM $\pi$ to perform atomic fact extraction (with prompt $p_{atomic\_fact}$) and natural language inference (with prompt $p_{nli}$). The hallucination score and critique feedback are obtained using the atomic facts that are not entailed.

a summary is faithful or not, and also to rank unfaithful summaries. Specifically, $\hat{y}$ is considered to be unfaithful if $s > 0$. On the other hand, the textual feedback $c$ from critique is used to guide the subsequent refinement process. In this work, we explore two strategies for designing the LLM critique component.

**Critique with binary feedback:** LLM is prompted to output a simple yes/no response indicating whether the summary contains any hallucinated information. Specifically, given an input document $x$ and an initial summary $\hat{y}$, the hallucination score $s$ of $\hat{y}$ is defined as the log-likelihood ratio of "yes" and "no" tokens:

$$s = \log \frac{\pi(yes|x, \hat{y}, p_{critique}^{bin})}{\pi(no|x, \hat{y}, p_{critique}^{bin})} \tag{1}$$

where $p_{critique}^{bin}$ denotes the prompt designed for the binary critique strategy. The textual feedback $c$ in this case is "*The summary is unfaithful.*" if $s > 0$, and "*The summary is faithful.*", otherwise.

**Critique with fine-grained feedback:** Inspired by the recent advancement in hallucination detection task (Min et al., 2023; Scirè et al., 2024; Yang et al., 2024; Wan et al., 2024), we design a three-stage process for LLM critique with fine-grained feedback. In the first stage, we decompose $\hat{y}$ into a list of atomic facts $\{f_1, f_2, ...\} \sim \pi(.|\hat{y}, p_{atomic\_fact})$, where $f_j$ represents a single piece of information in $\hat{y}$, and $p_{atomic\_fact}$ is the prompt for atomic fact extraction. In the second stage, we perform natural language inference evaluating the entailment of each $f_j$ with $x$ as the context: $z_j \sim \pi(.|x, f_j, p_{nli}) \in \{entailed, neutral, contradicted\}$, where $p_{nli}$ is the prompt for natural language inference. Finally, the hallucination score is calculated based on the percentage of the atomic facts that are not entailed:

$$s = \frac{|\{f_j|z_j \neq entailed\}|}{|\{f_j\}|}, \tag{2}$$

and the textual feedback $c$ includes all the atomic facts that are not entailed. Figure 2 provides an illustration of LLM critique with fine-grained feedback using an example.

### 3.3 PREFERENCE LEARNING

For preference learning, we adopt a variant of Direct Preference Optimization (DPO) Rafailov et al. (2024) that uses a Negative Log-Likelihood (NLL) regularization term which has been shown to mitigate the degeneration problem of DPO (Cho et al., 2025; Pang et al., 2024; Liu et al., 2024). The resulting DPO + NLL objective is given by:

$$\max_{\theta} E_{D_{pref}}[\log \sigma(\beta \log \frac{\pi_{\theta}(y_{chosen}|x)}{\pi(y_{chosen}|x)} - \beta \log \frac{\pi_{\theta}(y_{rejected}|x)}{\pi(y_{rejected}|x)}) + \alpha \log \pi_{\theta}(y_{chosen}|x)], \tag{3}$$

where $\theta$ denotes the parameters of the summarization LoRA adapter, $\beta$ is the scaling factor, and $\alpha$ controls the strength of the NLL term.

Table 1: Comparison of LLM critique strategies.

|  | MiniCheck | GPT4-Likert | GEval Coh. | GEval Consist. | GEval Flu. | GEval Rel. |
|---|---|---|---|---|---|---|
| XSum |  |  |  |  |  |  |
| Pretrained LLM | 0.701 | 4.16 | 4.04 | 4.43 | 2.99 | 4.18 |
| SCRPO, Binary feedback | 0.748 | 4.25 | 4.02 | 4.51 | 2.99 | 4.19 |
| SCRPO, Fine-grained feedback | **0.761** | **4.38** | **4.12** | **4.66** | 2.99 | **4.23** |
| CNNDM |  |  |  |  |  |  |
| Pretrained LLM | 0.715 | 4.45 | **4.04** | 4.71 | 2.99 | 4.21 |
| SCRPO, Binary feedback | 0.803 | 4.55 | 3.94 | 4.77 | 2.99 | 4.20 |
| SCRPO, Fine-grained feedback | **0.806** | **4.65** | 4.01 | **4.81** | 2.99 | **4.23** |
| SAMSum |  |  |  |  |  |  |
| Pretrained LLM | 0.437 | 4.17 | 4.49 | 4.57 | 2.97 | 4.54 |
| SCRPO, Binary feedback | 0.498 | 4.32 | 4.48 | 4.62 | 2.96 | 4.43 |
| SCRPO, Fine-grained feedback | **0.523** | **4.42** | **4.56** | **4.75** | 2.96 | **4.60** |

## 4 EXPERIMENTS

### 4.1 DATASETS AND EVALUATION METRICS

We evaluate our proposed method with three widely-used summarization benchmarks: XSum (Narayan et al., 2018), CNNDM (See et al., 2017) and SAMSum (Gliwa et al., 2019). Both XSum and CNNDM datasets contain news articles, while SAMSum dataset consists of casual conversations mimicking everyday chats among family and friends. For each dataset, we sample 10,000 documents from the official training set and use them as the input document set $\{x\}$ in our SCRPO framework.

Following previous works (Wan et al., 2025a; Wadhwa et al., 2024), we measure the faithfulness of the summaries using MiniCheck (Tang et al., 2024) and a GPT-4 Likert-style evaluation (Li et al., 2024b). Both metrics show high correlations with human judgments of faithfulness. To assess the general quality of the summaries, we also report GEval (Liu et al., 2023) results, which include four scores measuring coherence, consistency, fluency and relevance.

### 4.2 IMPLEMENTATION

In this work, we use Qwen2.5-7B-Instruct (qwe, 2025) as the pretrained base model for all the components in SCRPO framework, including initial summarization, LLM critique and LLM refinement. We use a LoRA adapter (Hu et al., 2022) with rank of 16 and an alpha of 32. For the summarization task, we instruct the model to generate a single-sentence summary for the input document. Preliminary results show that the instruction following capability of Qwen2.5-7B-Instruct is sufficiently strong, so more than 99% of the generated summaries satisfy the single-sentence requirement. When evaluating both pretrained and finetuned models on the summarization task, we use beam search decoding with a beam size of 5 for generating summaries.

### 4.3 LLM CRITIQUE STRATEGIES

In our first experiment, we evaluate the effectiveness of the two LLM critique strategies designed for the SCRPO framework. The results, summarized in Table 1, yield the following observations: (i) Both strategies lead to significant performance improvement in terms of faithfulness metrics (MiniCheck and GPT-4 Likert scores) when compared to the pretrained model. (ii) They also maintain or slightly improve the overall summary quality, as reflected in GEval scores. The only exception is that the critique with binary feedback strategy shows a small drop in GEval-Relevance for SAMSum dataset. (iii) The fine-grained feedback strategy achieves larger gains on faithfulness metrics. Overall, these findings highlight the importance of leveraging LLM critique and refinement abilities for faithful summarization, and demonstrate that the SCRPO framework can effectively self-distill a model's knowledge about faithfulness into its summarization ability. Based on these results, we adopt the fine-grained feedback strategy for SCRPO in all the subsequent experiments.

### 4.4 PREFERENCE TRIPLET SELECTION

In this experiment, we compare three preference triplet selection strategies within the SCRPO framework: (i) *Single beam search:* We generate a single initial summary $\hat{y}_{beam}$ using beam search. Then,

Table 2: Comparison of preference triplet selection strategies

|  | MiniCheck | GPT4-Likert | GEval Coh. | GEval Consist. | GEval Flu. | GEval Rel. |
|---|---|---|---|---|---|---|
| **XSum** | | | | | | |
| Pretrained LLM | 0.701 | 4.16 | 4.04 | 4.43 | 2.99 | 4.18 |
| SCRPO, Single beam search | 0.828 | 4.44 | 3.93 | 4.58 | 2.99 | 3.97 |
| SCRPO, Random selection | 0.748 | 4.34 | 4.04 | 4.61 | 2.99 | 4.17 |
| SCRPO, Extreme selection | 0.761 | 4.38 | 4.12 | 4.66 | 2.99 | 4.23 |
| **CNNDM** | | | | | | |
| Pretrained LLM | 0.715 | 4.45 | 4.04 | 4.71 | 2.99 | 4.21 |
| SCRPO, Single beam search | 0.876 | 4.73 | 3.78 | 4.76 | 2.99 | 4.03 |
| SCRPO, Random selection | 0.824 | 4.62 | 3.92 | 4.81 | 2.99 | 4.16 |
| SCRPO, Extreme selection | 0.806 | 4.65 | 4.01 | 4.81 | 2.99 | 4.23 |
| **SAMSum** | | | | | | |
| Pretrained LLM | 0.437 | 4.17 | 4.49 | 4.57 | 2.97 | 4.54 |
| SCRPO, Single beam search | 0.566 | 4.43 | 4.46 | 4.66 | 2.96 | 4.50 |
| SCRPO, Random selection | 0.528 | 4.33 | 4.49 | 4.67 | 2.97 | 4.53 |
| SCRPO, Extreme selection | 0.523 | 4.42 | 4.56 | 4.75 | 2.96 | 4.60 |

we critique and refine $\hat{y}_{beam}$ to generate a single refined summary $\hat{y}_{r,beam}$ by using beam search in both the steps. Finally, we form a preference triplet by using the initial summary $\hat{y}_{beam}$ as $y_{rejected}$, and the refined summary $\hat{y}_{r,beam}$ as $y_{chosen}$. (ii) *Random selection:* We repeat LLM summarization/critique/refinement steps several times to generate multiple initial unfaithful summaries and the corresponding refined summaries. Then, we randomly select one initial unfaithful summary as $y_{rejected}$, and one refined summary as $y_{chosen}$. (iii) *Extreme selection:* After generating multiple initial unfaithful summaries and the corresponding refined summaries, we choose the worst unfaithful summary based on the hallucination score as $y_{rejected}$ and the refined summary derived from the best unfaithful initial summary as $y_{chosen}$.

The results in Table 2 show that, while finetuning on beam search-based preference data leads to the best performance in terms of faithfulness metrics (MiniCheck and GPT4-Likert), it results in lower overall summary quality (reflected in GEval scores) when compared to finetuning on extreme selection-based preference data. Random selection strategy performs either similarly or worse when compared to the extreme selection strategy in majority of the cases except on MiniCheck score in the case of CNNDM and SAMSum datasets. Based on these results, we choose the extreme selection strategy as it provides a good balance between faithfulness and overall summary quality.

### 4.5 COMPARISON WITH ALTERNATIVE APPROACHES

In this experiment, we compare SCRPO with two previous state-of-the-art approaches, MPO (Choi et al., 2024) and SCOPE (Duong et al., 2025). Both MPO and SCOPE follow a two-stage framework: the first stage performs supervised fine-tuning (SFT) on data with human-annotated summaries, and the second stage further improves the model by training on a preference dataset generated by the first-stage model. Since we focus on self-supervised setting in this work, we omit the first SFT stage and instead use the pretrained LLM directly to construct the preference dataset for the second stage.

We further evaluate three alternative variants of SCRPO for comparison:

**SCRPO - Inference time** performs LLM critique and refinement with beam search decoding directly during inference.

**SCRPO - Critique only** takes an input document $x$ and a set of initial summaries, applies LLM critique with fine-grained feedback to assign hallucination scores, and selects the lowest- and highest-scoring summaries to form a preference pair without any refinement. This approach follows the spirit of the self-rewarding paradigm, but with a reward signal explicitly tailored for faithful summarization.

**SCRPO - SFT** constructs the self-generated preference dataset and then performs SFT on $D_{sft} = \{(x_i, y_{chosen})\}$, aligning the model's outputs with the refined summaries.

All the above variants of SCRPO and previous methods aim to enhance the faithfulness of LLM summarization in a fully self-supervised manner, without external knowledge resources or strong teacher models. Also, all these methods incur no additional computational cost at inference time except SCRPO - Inference time.

Table 3: Comparison of SCRPO with various alternative approaches. Bold font indicates the best performing method for each metric. We do not show any result in bold for a metric if all the methods perform equally.

| | MiniCheck | GPT4-Likert | GEval Coh. | GEval Consist. | GEval Flu. | GEval Rel. |
|---|---|---|---|---|---|---|
| **XSum** | | | | | | |
| Pretrained LLM | 0.701 | 4.16 | 4.04 | 4.43 | 2.99 | 4.18 |
| MPO (Choi et al., 2024) | 0.694 | 4.13 | 4.03 | 4.40 | 2.98 | 4.17 |
| SCOPE (Duong et al., 2025) | 0.713 | 4.14 | 4.06 | 4.42 | 2.99 | 4.18 |
| SCRPO - Inference time | 0.722 | 4.23 | 4.06 | 4.44 | 2.99 | 4.16 |
| SCRPO - Critique only | 0.738 | 4.33 | 4.11 | 4.62 | 2.99 | **4.24** |
| SCRPO - SFT | 0.735 | 4.23 | 4.04 | 4.49 | 2.99 | 4.18 |
| SCRPO | **0.761** | **4.38** | **4.12** | **4.66** | 2.99 | 4.23 |
| **CNNDM** | | | | | | |
| Pretrained LLM | 0.715 | 4.45 | **4.04** | 4.71 | 2.99 | 4.21 |
| MPO (Choi et al., 2024) | 0.712 | 4.42 | 4.03 | 4.70 | 2.99 | 4.23 |
| SCOPE (Duong et al., 2025) | 0.721 | 4.43 | 4.04 | 4.74 | 2.99 | 4.22 |
| SCRPO - Inference time | 0.746 | 4.48 | 4.00 | 4.73 | 2.99 | 4.22 |
| SCRPO - Critique only | 0.752 | 4.53 | 4.02 | 4.74 | 2.99 | 4.23 |
| SCRPO - SFT | 0.763 | 4.53 | 4.01 | 4.76 | 2.99 | 4.23 |
| SCRPO | **0.806** | **4.65** | 4.01 | **4.81** | 2.99 | 4.23 |
| **SAMSum** | | | | | | |
| Pretrained LLM | 0.437 | 4.17 | 4.49 | 4.57 | 2.97 | 4.54 |
| MPO (Choi et al., 2024) | 0.456 | 4.18 | 4.47 | 4.57 | 2.95 | 4.52 |
| SCOPE (Duong et al., 2025) | 0.440 | 4.17 | 4.48 | 4.55 | 2.96 | 4.51 |
| SCRPO - Inference time | 0.470 | 4.21 | 4.47 | 4.62 | 2.97 | 4.53 |
| SCRPO - Critique only | 0.487 | 4.32 | 4.55 | 4.71 | 2.98 | **4.61** |
| SCRPO - SFT | 0.498 | 4.27 | 4.52 | 4.68 | 2.97 | 4.59 |
| SCRPO | **0.523** | **4.42** | **4.56** | **4.75** | 2.96 | 4.60 |

The results in Table 3 lead to several important observations. (i) Prior methods, MPO and SCOPE, fail to consistently surpass the pretrained LLM in terms of faithfulness, suggesting that their preference data construction is not aligned with faithfulness as measured by MiniCheck and GPT4-Likert scores. (ii) The SCRPO - Inference time variant achieves higher faithfulness than the pretrained LLM while maintaining overall summary quality. While this approach does not require training data, it needs additional compute during inference. (iii) The proposed SCRPO framework outperforms prior state-of-the-art methods and all the SCRPO variants considered, delivering stronger results in terms of both faithfulness and overall quality. Notably, comparisons with SCRPO - Critique only and SCRPO - SFT underscore the importance of the refinement and preference learning components in our SCRPO framework.

### 4.6 Cross domain generalization ability

In this section, we investigate the cross-domain generalization ability of the SCRPO framework. Specifically, we perform SCRPO with input documents from a source domain and evaluate the faithfulness and overall quality of summaries generated for documents from a different target domain. We can make the following observations form the results shown in Table 4: (i) Applying SCRPO with a different source domain still outperforms the pretrained LLM. (ii) When the source and target domains are related (e.g., both XSum and CNNDM contain news articles), SCRPO continues to surpass the inference-time method. (iii) Even under substantial domain shifts (e.g., from news to conversational data), SCRPO demonstrates robustness, achieving marginal improvements over the inference-time method while retaining the benefit of computational efficiency. These results suggest that the benefits of using SCRPO extend beyond the training dataset.

### 4.7 Model size

The proposed SCRPO framework relies on the internal critique and refinement capabilities of LLMs. This raises a natural question: Can smaller models also use SCRPO and improve themselves? To answer this, we finetuned Qwen2.5 instruction-following models of different sizes (0.5B, 1.5B, 3B, and 7B) on XSum dataset using SCRPO. We compare the pretrained and finetuned models using MiniCheck and GPT4-Likert scores in Figure 3. While 3B and 7B models are able to self-improve, the 0.5B and 1.5B models experience significant degradation in faithfulness. These results suggest that a minimum model capacity is required for SCRPO to be effective.

Table 4: Cross domain generalization ability.

| | MiniCheck | GPT4-Likert | GEval Coh. | GEval Consist. | GEval Flu. | GEval Rel. |
|---|---|---|---|---|---|---|
| XSum | | | | | | |
| Pretrained LLM | 0.701 | 4.16 | 4.04 | 4.43 | 2.99 | 4.18 |
| SCRPO, inference time | 0.722 | 4.23 | 4.06 | 4.44 | 2.99 | 4.16 |
| SCRPO, cross domain (SAMSum → XSum) | 0.747 | 4.27 | 4.03 | 4.52 | 2.99 | 4.15 |
| SCRPO, cross domain (CNNDM → XSum) | 0.793 | 4.39 | 4.04 | 4.64 | 2.99 | 4.18 |
| SCRPO, target domain (XSum) | 0.761 | 4.38 | 4.12 | 4.66 | 2.99 | 4.23 |
| SAMSum | | | | | | |
| Pretrained LLM | 0.437 | 4.17 | 4.49 | 4.57 | 2.97 | 4.54 |
| SCRPO, inference time | 0.470 | 4.21 | 4.47 | 4.62 | 2.97 | 4.53 |
| SCRPO, cross domain (XSum → SAMSum) | 0.471 | 4.22 | 4.51 | 4.64 | 2.98 | 4.55 |
| SCRPO, target domain (SAMSum) | 0.523 | 4.42 | 4.56 | 4.75 | 2.96 | 4.60 |

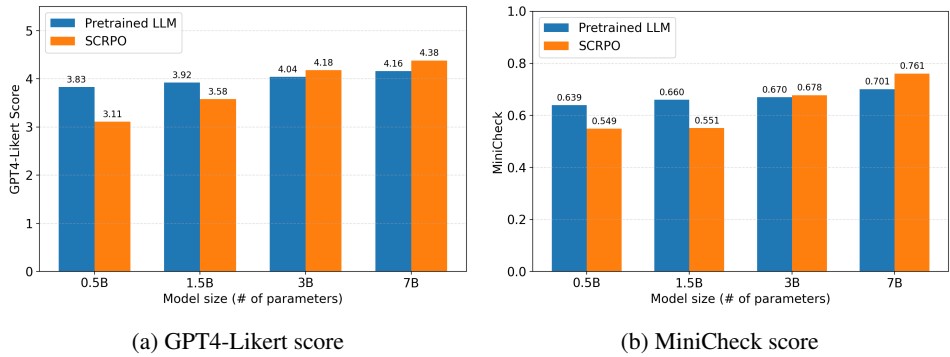

(a) GPT4-Likert score        (b) MiniCheck score

Figure 3: Impact of SCRPO training on models of different sizes (dataset - XSum).

## 4.8 HUMAN EVALUATION

We conducted a human evaluation study to comprehensively assess the benefits of the proposed SCRPO framework. We randomly sample 80 documents from the XSum test set and, for each document, generate two summaries: one produced by the pretrained LLM and the other by SCRPO. Six human annotators are recruited to perform two comparison-based eval-

Table 5: Human evaluation on faithfulness and general quality of summaries.

| | SCRPO wins | Tie | Pretrained LLM wins |
|---|---|---|---|
| Faithfulness | **24%** | 75% | 1% |
| General quality | 31% | 41% | 28% |

uation tasks: one focusing on selecting the more faithful summary, and the other on selecting the summary with better overall quality. The results in Table 5 illustrate that SCRPO clearly outperforms the pretrained LLM in faithfulness, while achieving comparable performance in overall quality. These results align closely with the trends observed from automatic evaluation metrics.

## 5 CONCLUSION

We introduce Self Critique and Refinement-based Preference Optimization (SCRPO), a novel framework that distills the critique and refinement capability of LLMs to improve their own faithful summarization. SCRPO achieves this by constructing a self-generated preference dataset and applying preference learning, enabling the model to enhance itself without external supervision. Extensive experiments demonstrate that SCRPO outperforms both prior state-of-the-art methods and its own variants in terms of faithfulness and overall summary quality. In particular, SCRPO surpasses its inference-time counterpart with the same self critique and refinement mechanism by delivering higher summary quality with greater inference-time efficiency. Moreover, SCRPO exhibits cross-domain generalization ability, underscoring its broad applicability.

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

## A  APPENDIX

### A.1  USE OF LARGE LANGUAGE MODELS (LLMs)

LLM served as a general-purpose writing assistant, helping refine the wording, grammar, and overall readability of the text. It did not contribute to the research ideas, methodology, experiments, analysis, or conclusions. Authors take full responsibility for the scientific content of this paper.

**Summarization** ($p_{summ}$)

Document:

[Document]

Please write a brief summary for the given document. The summary should be one sentence.

**Binary judgment** ($p_{judge}^{bin}$)

Below is a document and a corresponding summary. Please determine whether the summary contains hallucinated information that is not supported by the document.

Document:

[Document]

Summary:

[Summary]

State the final answer exactly as either 'Yes' (if hallucinated information is found) or 'No' (if not). Do not provide any additional information.

**Atomic fact extraction** ($p_{atomic\_fact}$)

Given the following sentence, list all simple facts it contains. Each fact should be a minimal statement that expresses a single piece of information. Each fact must be written so it makes sense by itself, without relying on the context.

Sentence: [Sentence]

Answer in the following format:

Facts:

1.

2.

...

**Natural language inference** ($p_{nli}$)

Given the context, determine if the statement is entailed or contradicted or neutral.

Context: [Context]

Statement: [Statement]

Answer with "Entailed", "Contradicted" or "Neutral"

**Refinement** ($p_{refine}$)

You will be given a document, a summary, and comment on the summary. Your task is to revise the summary given the comment. Please make sure you address all the suggestions by only making the least amount of changes.

Document: [Document]

Summary:

[Summary]

Comment:

[Comment]

Please check the document for the correct information and make appropriate edits.

Table 6: Prompt templates.

### A.2 PROMPT TEMPLATES

Table 6 shows the prompts we use for different components of SCRPO framework.

### A.3 HUMAN EVALUATION INSTRUCTIONS

We adopt the human evaluation method proposed in previous works (Choi et al., 2024; Duong et al., 2025) with minor modifications. The detailed design of the human evaluation instructions are illustrated in Table 7.

### A.4 EXAMPLES OF SUMMARIZATION RESULTS

In Table 8, we present two examples of input documents from WikiNews dataset [1], and corresponding summaries produced by various methods, including pretrained LLM, test-time refinement method, and the proposed SCRPO. The unfaithful contents are highlighted in blue.

---

[1] https://www.wikinews.org

**Faithfulness evaluation**

Your task is to assess which summary is more faithful to the corresponding document. In this context, a summary is considered faithful if all information it contains is directly supported by the content of the document.
* If the summary introduces any unsupported or incorrect information, it should be rated as unfaithful.
* If both descriptions contain one or more faithfulness issues, rate them as a Tie.

To guide your evaluation:
* Carefully compare each detail in the summary with the document to ensure accuracy.
* A summary should not distort or add information that is not present in the document.
* If you notice even a single instance of unsupported information in a summary, it should be rated as unfaithful.
* If both descriptions have one or several faithfulness issues, they should both be considered unfaithful and rated as 'Tie'.

Please choose between the following options for each comparison:
* Summary A is more faithful
* Summary B is more faithful
* Tie (if both summaries are equally faithful or contain faithfulness issues)

Document:
{{Document}}

Summary A:
{{SummaryA}}

Summary B:
{{SummaryB}}

Please type "A", "B", or "Tie", to provide the answer.

**General quality evaluation**

Which of the following summaries does a better job of summarizing the most important points in the given document, without including unimportant or irrelevant details? A good summary is both precise and concise.

Document:
{{Document}}

Summary A:
{{SummaryA}}

Summary B:
{{SummaryB}}

Please type "A", "B", or "Tie", to provide the answer.

Table 7: Human Evaluation Instructions for Faithfulness and General Quality.

| Input Document 1 |
| --- |
| Mexican candidate confident of winning OAS leadership race |
| Mexican foreign minister Luis Ernesto Derbez says he is confident that he will become the next Organization of American States (OAS) Secretary General, and will be in the post within 4 months. He claims to have 17 of the 18 votes needed in order to have the support of a majority within the 34 member states of the OAS.
The post became available after the previous Secretary General, Miguel Angel Rodriguez, served only 24 days of his term of office, before returning home to Costa Rica to face corruption charges.
The other candidates for the post are ex El Salvadorean president Francisco Flores, Guatemalan Rigoberta Menchu and Chilean Interior Minister Jose Miguel Insulza. Meanwhile Francisco Flores is in the Dominican Republic on the beginning of a tour of Caribbean countries drumming up support for his own candidacy. He is being accompanied by El Salvador's deputy foreign minister Margarita Escobar in a clear sign of the importance El Salvador puts on getting their candidate into the job. |
| **Summary 1, Pretrained model** |
| Mexican foreign minister Luis Ernesto Derbez is confident he will become the next Secretary General of the Organization of American States within 4 months, claiming to have the support of 17 out of 18 member states, while other candidates include Francisco Flores from El Salvador, Rigoberta Menchu from Guatemala, and Jose Miguel Insulza from Chile. |
| **Summary 1, inference-time method** |
| Mexican foreign minister Luis Ernesto Derbez is confident he will become the next Secretary General of the Organization of American States within 4 months, claiming to have the support of 17 out of 18 member states, while other candidates include Francisco Flores from El Salvador, Rigoberta Menchu from Guatemala, and Jose Miguel Insulza from Chile. |
| **Summary 1, SRPO (ours)** |
| Mexican foreign minister Luis Ernesto Derbez is confident he will become the next Secretary General of the Organization of American States (OAS) within 4 months, claiming to have 17 of the 18 votes needed, while other candidates include Francisco Flores from El Salvador, Rigoberta Menchu from Guatemala, and Jose Miguel Insulza from Chile. |
| **Input Document 2** |
| Doc:Results of the 2005 New Zealand General Election |
| Labour- has won 50 seats and National Party has won 49 in the 122-member House of Representatives , which has increased by two (from 120) because the Maori Party won four electorate seats, more than its proportion of the party vote entitled it to (Called in MMP an overhang).
Neither the Labour-led coalition or the National Party have won sufficient seats to govern on their own, and will require the backing of other smaller parties to govern in a coalition.
There are 193,348 special votes still to be counted. Official results are expected to be announced on 1 October 2005.
Preliminary election results in depth
The preliminary results for the 2005 General Election are from the Chief Electoral Office website [1] plus TV coverage (One, and TV3)
The official results will be released after inclusion of special votes and re-counting of all votes over the coming two weeks.
Results by party |
| **Summary 2, Pretrained model** |
| The 2005 New Zealand General Election resulted in a Labour Party victory with 50 seats, one seat short of a majority, while the National Party won 49 seats, and both parties will need support from smaller parties to form a government under the Mixed Member Proportional (MMP) voting system. |
| **Summary 2, Inference-time method** |
| The 2005 New Zealand General Election resulted in a tie between the Labour Party and the National Party, each winning 50 seats and 49 seats respectively in the 122-member House of Representatives. Neither party won sufficient seats to govern on their own and will require the backing of other smaller parties to form a government under the Mixed Member Proportional (MMP) voting system. |
| **Summary 2, SRPO (ours)** |
| The 2005 New Zealand General Election resulted in the Labour Party winning 50 seats and the National Party winning 49 seats in the 122-member House of Representatives, with neither party having enough seats to govern on their own and requiring the backing of other smaller parties to form a coalition. |

Table 8: Summarization results from our method and baselines. The unfaithful contents are highlighted in blue.

