# OpenReview forum: "Learning from Self Critique and Refinement for Faithful LLM Summarization"
_ICLR.cc/2026/Conference — ICLR 2026 Conference Withdrawn Submission_

### Official Review · Reviewer_WLTS · 2025-10-27

**Soundness:** 3
**Presentation:** 4
**Contribution:** 3
**Rating:** 8
**Confidence:** 4

**Summary:**

The paper introduces Self-Critique and Refinement-based Preference Optimization (SCRPO), a self-supervised technique for improving faithfulness with summarization models. Under this technique, a pre-trained LLM receives a set of unlabelled documents and generates pairs of summaries (preferred, not preferred) via critiquing and refining its own outputs (the authors experiment with 2 critiquing signals - binary and fine-grained). After that, the model is preference-tuned on the resulting pairs, which improves faithfulness of the model.

The authors evaluate the technique on 2 summarization datasets: XSum, CNNDM and SAMSum. Output summaries are evaluated synthetically (MiniCheck GPT-4 Likert-style evaluation and GEval) and using human judges. The authors use Qwen-2.5-7B-Instruct base model and compare the performance of it off-the-shelf, several variations of SCRPO training as well as alternative techniques - MPO and SCOPE. Evaluation results show consistent improvement against the metrics after SCRPO training. Ablation studies on model size shows that SCRPO helps improve faithfulness to medium-sized models (3B, 7B) and hurts smaller models (0.5B, 1.5B)

**Strengths:**

* a novel framework is proposed for improving faithfulness of summaries in a completely self-supervised way
* SCRPO outperforms base models and alternative techniques, even those not completely self-supervised

**Weaknesses:**

* the effectiveness of the technique is limited by the model's self-critique ability - which is demonstrated in the model size ablation study where using SCRPO actually hurts smaller models' (0.5B, 1.5B) performance
* given the point above, an experiment with a stronger teacher model (e.g. GPT / Claude) as preference pair generator would provide a fuller picture of the evaluation. That would show how much reliance on the model's own judgements limits the performance of the main setup evaluated, Qwen@7B

**Questions:**

N/A

---

### Official Review · Reviewer_got5 · 2025-10-29

**Soundness:** 2
**Presentation:** 4
**Contribution:** 3
**Rating:** 4
**Confidence:** 4

**Summary:**

This paper addresses the issue of mitigating hallucinations in LLM outputs for abstractive summarization, without relying on more capable teacher models or requiring additional test-time computation. The authors propose SCRPO, a framework designed to enable self-improvement in the target LLM. Given unlabeled documents, SCRPO prompts the LLM to generate initial summaries and then collects critiques in the form of either binary feedback or fine-grained feedback. Preference pairs are then selected based on hallucination scores for preference learning using a variant of the DPO algorithm.

Experiments are conducted using Qwen2.5-7B on multiple datasets, including CNN/DM, XSum, and SAMSum. The results demonstrate that SCRPO achieves consistent improvements across different datasets compared to the vanilla model and two baselines (MPO and SCOPE). Further analysis confirms the effectiveness of the design choices: fine-grained feedback leads to greater gains in faithfulness metrics, while extreme selection strikes a better balance between faithfulness and overall summary quality. Additionally, SCRPO exhibits cross-domain generalization capabilities, performing competitively with or even surpassing its inference-time variant.

**Strengths:**

* The paper is well-motivated and features a clear, easy-to-follow flow of presentation.
* The proposed approach, SCRPO, is intuitive in the context of related work and is thoroughly discussed through various design choices and ablation studies.
* SCRPO demonstrates consistent improvements across different benchmarks in terms of both faithfulness metrics and the overall summary quality reflected by automatic evaluations.

**Weaknesses:**

* Although the improvements across benchmarks are consistent, the findings are not fully convincing, since:
	* the paper does not mention whether the results are stable over multiple runs or supported by significance testing;
	* the approach was only evaluated on Qwen2.5, leaving its effectiveness on other backbone LLMs unverified.

* The human evaluation remains limited, as it does not include results on CNN/DM or SAMSum or comparisons with other baselines, nor does it report inter-annotator agreement.

* More in-depth analysis is expected, such as:
	* whether the approach remains effective with larger models and can genuinely advance the state-of-the-art performance;
	* the relationship between the minimum required model capacity and task complexity;
	* which kind of hallucinations remain unmitigated.

**Questions:**

* The relationship between hallucination severity trends and the effectiveness of different approaches across varying model scales.
* The correlation between the minimum effective model size and task difficulty.
* Human evaluation comparing SCRPO with MPO and SCOPE.
* Performance validation using other backbone LLMs.
* Error analysis identifying which types of hallucinations remain unmitigated by self-improvement methods.

---

### Official Review · Reviewer_YcDK · 2025-10-30

**Soundness:** 3
**Presentation:** 3
**Contribution:** 2
**Rating:** 4
**Confidence:** 3

**Summary:**

This paper proposes a framework, SCRPO, to improve the faithfulness of LLM summarization quality by constructing a preference dataset using LLM generation and self-critique, and performing preference learning over it. To construct the preference dataset, the proposed framework first generates summaries from documents, and evaluates each summary using hallucination scores. Then it asks the LLM to refine the summaries for several rounds, and uses the refined summary with the lowest hallucination score vs. the initial summary as the chosen vs. rejected preference pairs. It then performs preference learning over the constructed dataset. Experiments were performed over three datasets, XSum, CNN/DM and SAMSum. Results show that the proposed framework attains consistent improvement over the baseline.

**Strengths:**

* The proposed method is technically sound and intuitively reasonable. The design is lightweight and clean.
* The performance gain is consistent across datasets and metrics.
* Ablation and analysis are extensive, exploring various aspects relating to the design choices of the proposed framework.

**Weaknesses:**

* In the ablation on model size, SCRPO decreases the faithfulness for the 0.5B and 1.5B models, and achieves only marginal improvement on the 3B model. This raises some concerns about the general applicability of SCRPO. Related to this, it would be useful to include results on larger models, as demonstrating effectiveness across mid- and large-scale models would suggest broader generalizability while restrictions to specific model size bands would indicate limited scope.
* The technical novelty of this paper is somewhat limited, as there has already been a variety of works on using preference learning from LLM’s own generations to improve faithfulness/factuality (MPO and SCOPE cited and compared in the paper, and works like [1] for summarization and [2,3,4] for general open-ended generation). In this sense, performing preference learning for factuality based on self-generated outputs, and expecting corresponding performance improvements, is not particularly uncommon. The design of the hallucination score also generally follows established paradigms like FactScore [5]. Compared to existing works, the new insight of this paper seems to be that constructing preference pairs from the refined vs. original generated summaries is a better strategy than from 2 distinct generated summaries (SCRPO vs. Critique only). This finding is interesting but its contribution seems limited as a standalone insight. Also, since this comparison is performed only for one base model, it is difficult to assess how broadly it holds. For example, in settings like the 3B model experiment, where the gap between the baseline and SCRPO is already small, it is unclear whether SCRPO would still maintain an advantage over alternative designs.
* It would be helpful to provide more analysis on the hallucination score side, such as its distribution for accepted vs. rejected summaries, how accurate and reliable they are, and how their reliability impacts preference learning (see Questions below).

[1] Song, Hwanjun, et al. "Learning to Summarize from LLM-generated Feedback." NAACL 2025.

[2] Tian, Katherine, et al. "Fine-tuning language models for factuality." ICLR 2023.

[3] Lin, Sheng-Chieh, et al. "Flame: Factuality-aware alignment for large language models." NeuRIPS 2024.

[4] Gu, Yuzhe, et al. "Mask-dpo: Generalizable fine-grained factuality alignment of llms." ICLR 2025.

[5] Min, Sewon, et al. "Factscore: Fine-grained atomic evaluation of factual precision in long form text generation." EMNLP 2023.

**Questions:**

* Ablation study on model size shows that applying the proposed SCRPO framework reduces summary faithfulness. What are the causes behind this? Is it that smaller models produce less accurate hallucination scores, or is it because the quality of the refined summaries are low?
* Also on the model size ablation, what about the larger models? It would be helpful to provide a band of the model sizes where the proposed method is effective, to clarify its scope and applicability.
* How many refinement rounds are needed? It would be helpful to perform some ablation on this, to see the cost-performance tradeoff.
* What do hallucination score distributions look like for the accepted vs. rejected summaries? Also, it would be helpful to provide some analysis or explanation on how accurate and reliable the hallucination scores are, and how they impact the performance of preference learning.

---

### Official Review · Reviewer_XZT5 · 2025-11-01

**Soundness:** 2
**Presentation:** 2
**Contribution:** 2
**Rating:** 4
**Confidence:** 3

**Summary:**

This paper proposes Self-Critique and Refinement-based Preference Optimization (SCRPO), a self-supervised training framework designed to enhance the faithfulness of LLM summarization. The method leverages an LLM’s own critique and refinement abilities to generate preference pairs — contrasting initial summaries and refined ones — and subsequently applies Direct Preference Optimization (DPO) with an additional NLL regularization term to fine-tune the model.

The authors further include ablation studies (critique strategies, triplet selection, cross-domain generalization, and model size analysis), and a human evaluation comparing SCRPO vs. the pretrained model.

**Strengths:**

1. Integrates self-critique and refinement into preference optimization.

2. Eliminates inference-time overhead while retaining refinement benefits.

3.Well-structured exposition, strong figures, and detailed appendices with prompts.

**Weaknesses:**

1. Human evaluation only contrasts SCRPO with the pretrained model, not with other baselines (e.g., SCOPE, MPO). This limits the strength of claims about outperforming state-of-the-art systems in human preference.

**Questions:**

Why was human comparison restricted to SCRPO vs. pretrained LLM? Would including SCOPE or MPO be feasible to better support the claim of SOTA human-level improvement?

---

### Note · Authors · 2025-12-02

I have read and agree with the venue's withdrawal policy on behalf of myself and my co-authors.